# Adaptive Model Pruning in Federated Learning through Loss Exploration

Christian Internò [1]   Elena Raponi [2]   Niki van Stein [2]   Thomas Bäck [2]   Markus Olhofer [3]   Yaochu Jin [4]
Barbara Hammer [1]

## Abstract

The rapid proliferation of smart devices coupled with the advent of 6G networks has profoundly reshaped the domain of collaborative machine learning. Alongside growing privacy-security concerns in sensitive fields, these developments have positioned federated learning (FL) as a pivotal technology for decentralized AI training. However, FL faces significant hurdles, including high communication overheads, computational limitations, and the complexities arising from non-IID data distributions. We propose `AutoFLIP`, a novel approach designed to enhance the scalability and efficiency of FL. `AutoFLIP` introduces a federated loss exploration phase that adaptively prunes non-essential model parameters. This process leverages gradient behavior insights across diverse client losses to optimize resource usage and computational load, crucial for training neural networks effectively at scale. Extensive experiments across various datasets and task demonstrate that `AutoFLIP` not only accelerates convergence but also significantly reduces computational and communication costs—by 48.8% and 35.5%, respectively—while achieving robust performance. These advances make `AutoFLIP` a possible solution for deploying efficient and scalable FL in various real-world applications, such as healthcare to smart cities.

## 1. Introduction

The proliferation of smart devices at the network edge, coupled with advancements in 6G networks, has created a decentralized setting. Multiple participants store their data locally, which offers an opportunity for collaborative model training, enhancing robustness and generalization. Distributing the computational load across these devices results in faster training times and lower energy consumption compared to centralized approaches. However, distributed Machine Learning (ML) faces significant challenges. Efficient communication and coordination among participants are crucial, as each device holds only a subset of the data. This requires designing algorithms that minimize data exchange while ensuring high-quality model convergence. Device heterogeneity, including differences in computational power, storage, and bandwidth, further complicates distributed training. Algorithms must adapt to such environments to scale up distributed learning. Privacy and security concerns, along with regulations like the European GDPR (htt, 2016), add another layer of complexity (Hoffpauir et al., 2023). With sensitive data distributed across various devices, ensuring the privacy of individual data points becomes essential. For example, medical data stored in hospitals and personal devices is valuable for training diagnostic models but is also subject to strict privacy and security regulations.

In this context, Federated Learning (FL) (Zhu et al., 2021) emerges as an effective strategy for training always more complex DL models while preserving the privacy of the data. FL facilitates collaborative model training across multiple devices without exposing local data. A central server, i.e., a global model, coordinates this process by aggregating the updates from locally trained models, which ensures a secure learning environment. Current FL research focuses on enhancing privacy and adapting ML workflows for specific uses, often with predetermined ML model configurations. Tasks related to computer vision may involve well-known neural network (NN) architectures like VGG-16 (Simonyan & Zisserman, 2015) (138 million parameters) or ResNet-50 (He et al., 2015) (25.6 million parameters). However, these complex networks risk overfitting, especially with small training data sizes. FL systems typically expect clients to have high-speed processors and sufficient computational power for local calculations and parameter updates. Yet, many edge devices, such as smartphones, wearables, and sensors, have limited computing and memory capacities, posing a challenge to DL model training systems (Hoffpauir et al., 2023). Additionally, communicating DL models with

---
[*]Equal contribution  [1]CITEC, University of Bielefeld, Germany [2]LIACS, Leiden University, Netherlands [3]Honda Research Institute EU, Germany [4]Westlake University, China. Correspondence to: Christian Internò <christian.interno@uni-bielefeld.de>, Elena Raponi <e.raponi@liacs.leidenuniv.nl>.

Accepted to the Workshop on Advancing Neural Network Training at International Conference on Machine Learning (WANT@ICML 2024).

millions of parameters presents significant obstacles for FL transmission (Shlezinger et al., 2020; Asad et al., 2023). Therefore, using FL effectively with edge devices that have limited computational capabilities, while maintaining efficient communication, remains an active research question. FL's effectiveness is further hindered by the prevalence of non-IID data in real-world scenarios (Zhu et al., 2021; Karimireddy et al., 2020). Non-IID data refers to the unique statistical properties of each client's dataset, reflecting their varied environments. This creates conflicting training goals for local and global models, leading to convergence towards different local optima. As a result, client model updates become biased, impeding global convergence (Zhu et al., 2021; Karimireddy et al., 2020). These challenges underscore the need for personalized and innovative approaches in FL, particularly in optimizing and compressing models to improve inference time, communication cost, energy efficiency, and complexity, all while maintaining satisfactory accuracy.

**Our contribution.** We introduce a novel automated federated learning approach via informed pruning (`AutoFLIP`), which uses a novel loss exploration mechanism to automatically prune and compress DL models. In our assumed single-server architecture, each client operates on the same initial deep NN structure that automatically prunes itself at each round, based on the extraction of shared knowledge for an informed model compression. Specifically, by analyzing the variability of weights during a local exploration phase, which provides insights into gradient behaviors on the loss landscapes across clients, and subsequent information aggregation, the Deep Learning (DL) models involved in a FL round are pruned automatically. This strategy allows for dynamically reducing the complexity of the models in FL environments, thereby optimizing performance with limited computational resources at the client level. With our experiments over various datasets, tasks, and realistic non-IID scenarios, we provide strong evidence of the effectiveness of `AutoFLIP`.

**Reproducibility.** Our code for reproducing the experiments is available on GitHub.[1]

## 2. Background and Related Work

**Pruning in Deep Learning.** Following the assumption that a DL model can contain a sub-network that represents the performance of the entire model after being trained, model pruning is a good strategy to reduce computational requirements of resource-constrained devices (Mozer & Smolensky, 1988a; LeCun et al., 1989; Janowsky, 1989). Most pruning approaches balance accuracy and sparsity during

---

[1] https://github.com/ChristianInterno/AutoFLIP

the inference stage by calculating the importance scores of parameters in a well-trained NN and removing those with lower scores. These scores can be derived from weight magnitudes (Janowsky, 1989; Han et al., 2015), first-order Taylor expansion of the loss function (Mozer & Smolensky, 1988b; Molchanov et al., 2017), second-order Taylor expansion (LeCun et al., 1989; Hassibi & Stork, 1992; Molchanov et al., 2019), and other variants (Louizos et al., 2018; Singh & Alistarh, 2020).

Another recent research direction in NN pruning focuses on improving training efficiency, divided into two categories: pruning at initialization and dynamic sparse training. Pruning at initialization involves pruning the original full-size model before training based on connection sensitivity (Lee et al., 2019), Hessian-gradient product (Wang et al., 2020), and synaptic flow (Tanaka et al., 2020). However, since this method does not involve training data, the pruned model may be biased and not specialized for the task. Dynamic sparse training iteratively adjusts the pruned model structure during training while maintaining the desired sparsity (Dettmers & Zettlemoyer, 2019; Evci et al., 2020). This approach requires memory-intensive operations due to the large search space, making it impractical for resource-constrained devices.

Initial attempts to use pruning for deploying deep neural networks on resource-limited devices have utilized pre-trained CNNs in a centralized setting (You et al., 2019; Lin et al., 2020). However, this approach can lead to reduced data privacy, higher costs, poor adaptation to local conditions, suboptimal performance on diverse data, and latency in real-time applications.

**Pruning in Federated Learning.** The widely accepted FL standard is known as `FedAvg` (McMahan et al., 2023). It distributes a global model to clients for local training and aggregates it by averaging their parameters. Empirical studies have shown the robustness of this approach, even when handling non-convex optimization problems (Das et al., 2022). As a result, it is commonly used as a standard for evaluating newly developed FL protocols. In this study, we will compare the performance of the proposed `AutoFLIP` method to FedAvg, with different State-of-the-Art (SotA) FL pruning approachs, as tested in (Wu et al., 2023). In fact, since data remains locally stored and cannot be shared, traditional centralized pruning approaches that rely on access to training data are not feasible in FL.

In the context of FL, there has been work focused on dynamic active pruning to increase communication efficiency during training. Liu et al. (2022); Zhou et al. (2021) introduced a method where pruning decisions are made dynamically based on the model's real-time performance evaluation, which significantly reduces the data exchanged during training but adds computational complexity to client devices.

Jiang et al. (2023) introduced `PruneFL`, a FL method that incorporates adaptive and distributed parameter pruning. Their approach utilizes an unstructured method that does not take advantage of the collective insights of participating clients to develop a cooperative structured pruning strategy. This is in contrast to the objectives of `AutoFLIP`, which seeks to harness client-specific knowledge to facilitate a structured approach to pruning. Lin et al. (2022) introduced a novel approach for adaptive per-layer sparsity, however without incorporating any parameter aggregation scheme to reduce the error caused by pruning. This challenge was addressed by Tingting et al. (2023) by moving the pruning process to the global model that works on a computationally more powerful server. The pruned model is distributed to each client, where it undergoes training. Subsequently, each client sends back to the server only the updated parameters, restoring the full structure of the model at the server. Although this study includes various parameter selection criteria from the literature, its pruning method does not incorporate the information gathered during model training. This contrasts with our strategy, `AutoFLIP`, which leverages such information to enhance the pruning process. Yu et al. (2023a) proposed Resource-aware Federated Foundation Models, focusing on integrating large transformer-based models into FL, with the limitation of not exploring other architectures. Our method, `AutoFLIP`, diverges by introducing a pruning strategy that avoids the need for continuous evaluation of parameter significance and is applicable across various FL aggregation algorithms and model architectures.

## 3. Preliminaries

We start by defining the notation we use and presenting the standard formulation of FL, and then introduce the problem definition and objective.

**Notation:** We consider a total number of $C$ clients. At each FL round, $K$ clients are chosen and trained on different batches of size $B$ for $E$ epochs. The total number of rounds is $R$, which represents our termination criterion. For the exploration phase, we denote with $C_{\text{exp}}$ the number of clients selected, which, in this study, we take as the totality $C$ of available clients. The exploration lasts for $E_{\text{exp}}$ epochs.

### 3.1. Federated Learning

In the conventional FL setting, each client $i$ ($1 \leq i \leq K$) possesses its own data distribution $p_i(x, y)$, where $x \in \mathbb{R}^d$ represents the $d$-dimensional input vector and $y \in \{1, \ldots, M\}$ is the corresponding label from $M$ classes. Each client has a dataset $D_i$ with $N_i$ data points: $D_i = \{(x_i^{(1)}, y_i^{(1)}), \ldots, (x_i^{(N_i)}, y_i^{(N_i)})\}$. It is assumed that in a non-IID scenario the data distribution $p_i(x, y)$ varies across clients. These data distributions $p_i(x, y)$ are sampled from a family $\mathcal{E}$ of distributions. The objective is for the clients

to collaboratively train a global model with parameters $W_{\text{global}}$, which will perform predictions on new data. The global loss function for a data point $(x, y)$ is denoted by $\mathcal{L}(W_{\text{global}}, x, y)$, where the global objective function to be minimized is defined as:

$$\mathcal{L}(W_{\text{global}}) := \frac{1}{C} \sum_{i=1}^{C} \mathbb{E}_{(x_i, y_i) \sim p_i} [\mathcal{L}(W_{\text{global}}, x_i, y_i)], \quad (1)$$

with $\mathbb{E}_{(x_i, y_i) \sim p_i}$ representing the expected loss over the data distribution $p_i$ for each client $i$ with parameters $W$.

The optimization process involves several key steps:
**1. Client Selection:** A subset of $K$ clients is selected from the total $C$ clients.
**2. Local Training**: Each selected client $i$ performs local training for $E$ epochs using its local dataset $D_i$. The local training aims to minimize the local objective function $\mathcal{L}(W_i)$ using stochastic gradient descent (SGD): let $W_i^r$ be the local model parameters of client $i$ at round $r$, the update rule is given by: $W_i^{r+1} = W_i^r - \eta \nabla \mathcal{L}(W_i^r)$, where $\eta$ is the learning rate.
**3. Parameter Aggregation:** After local training, each client sends its updated parameters $W_i^{r+1}$ to the central server. The server aggregates these parameters to form the new global model $W_{global}^{r+1}$ using a weighted average: $W_{global}^{r+1} = \frac{1}{K} \sum_{i=1}^{K} W_i^{r+1}$. This iterative process is repeated for $R$ rounds the termination criterion is met.

### 3.2. Problem Definition and Objective

The non-IID nature of the data introduces challenges in ensuring the global model effectively generalizes across diverse client data distributions. The variance $\sigma_{\Delta W}^2$ of the weight updates received by the global model, as discussed by Zhu et al. (2021), can introduce noise and bias trajectories that potentially slow convergence and affect global model accuracy. Our objective with `AutoFLIP` is:

**i)** Minimize $\sigma_{\Delta W}^2$ through an informed adaptive pruning strategy across clients to mitigate noise and biases into the client trajectory, which could slow the global model convergence. This approach aims to reduce the impact of divergent client learning paths caused by non-IID data.

**ii)** Enhance training and communication efficiency. Pruning reduces the number of parameters that need to be communicated between clients and the server, thus lowering the communication overhead and expediting the overall training process.

## 4. Methodology

`AutoFLIP` is an automatized FL approach that utilizes informed pruning through a federated client loss exploration

process. Inspired by the idea of utilizing agents with similar tasks as *scouts* which explore the conformation of different loss function landscapes from Nikolić et al. (2023), `AutoFLIP` introduces a preliminary step to the FL rounds, which we term *federated loss exploration* phase. Here, a $C_{\exp}$ portion of clients (or the totality $C$), which inherit their model structure from the global model, explore for a number of $E_{\exp}$ exploration epochs their loss landscape using its local dataset $D_i$. Based on this, for each client $c_{\exp_i}$, we compute a local guidance matrix $G_{\text{local}_i}$, which records how important a certain parameter $W_i$ (weight or bias) is in terms of parameter deviation, i.e., change in parameter value after the exploration from its Xavier normal initialization (Kumar, 2017) value, which reflects loss variability. Afterward, we aggregate the information collected locally in a global pruning guidance matrix $PG_{\text{global}}$ on the server, which will generate an informed pruning mask to guide the pruning of the client models. The pruning workflow of `AutoFLIP` is illustrated in Figure 1. Please note that the initial federated loss exploration, computation of parameter deviations, and definition of local guidance matrices occur only once at the beginning of the FL optimization process as a preliminary procedure. In contrast, the global guidance matrix and subsequent pruning strategy are automatically redefined in each FL round, considering the clients participating in that round.

To summarize, the iterative procedure consists of **(1)** pruning local models using the updated pruning guidance matrix, **(2)** training the pruned local models, **(3)** aggregating the locally trained model parameters and **(4)** evaluating performance and updating the pruning guidance matrix in each FL round.

### 4.1. Federated Loss Exploration

In `AutoFLIP`, the model initialization phase is augmented by a crucial federated loss exploration phase, allowing clients to explore their loss function landscapes. We envision each client as an explorer delving into different regions of their loss landscape. Through this exploration, they can identify crucial dimensions and those that can be disregarded based on their experience by quantifying gradient variability during the exploration. Subsequently, they transmit this knowledge to the server, which updates a pruning guidance mask $PG_{\text{global}}$. This mask is then shared among participating clients in each FL round to guide the evolution of client model structures within an informed pruning session.

To construct the mask $PG_{\text{global}}$, we begin with an initial exploration phase conducted on $C_{\exp}$ clients. In this study, we consider $C_{\exp} = C$. In our study we let explore the clients for $E_{\exp} = 150$ epochs, and for each model parameter we evaluate its evolution in the search space during the loss exploration. This evaluation is conducted by calculating the deviation $D_{i,m}$ for the $m^{th}$ parameter of a client model $i$ as

the squared difference between the initial ($W_{i,m}^{\text{Initial}}$) and final ($W_{i,m}^{\text{Final}}$) parameter values after $E_{\exp}$ epochs of exploration:

$$D_{i,m} = (W_{i,m}^{\text{Initial}} - W_{i,m}^{\text{Final}})^2. \qquad (2)$$

Using stochastic gradient descent for exploration, the deviation $D_{i,m}$ in Eq. (2) serves as a measure of gradient variability on the loss landscape for parameter $m$ during the preliminary exploration phase before the actual FL procedure. The greater the variation in the parameter space, the faster the improvements in loss: the update rule for a parameter in stochastic gradient descent is $W_{i,m}^{(e_{\exp}+1)} = W_{i,m}^{(e_{\exp})} - \eta \nabla \mathcal{L}_i \left( W_{i,m}^{(e_{\exp})}; D_i \right)$, where $W_{i,m}^{(e_{\exp})}$ and $W_{i,m}^{(e_{\exp}+1)}$ are the values of the parameter $m$ at the exploration epochs $e_{\exp}$ and $e_{\exp} + 1$, $\eta$ is the learning rate, and $\nabla \mathcal{L}_i \left( W_{i,m}^{(e_{\exp})} \right)$ is the gradient of the loss function of client $i$ with respect to the parameter $m$ at epoch $e_{\exp}$ using its local dataset $D_i$. Given the gradient update rule, the deviation in $W_{i,m}$ from the initial to the final exploration epoch $E_{\exp}$ can be approximated to $W_{i,m}^{\text{Final}} - W_{i,m}^{\text{Initial}} \approx -\eta \sum_{t=1}^{E_{\exp}} \nabla \mathcal{L}_i \left( W_{i,m}^{(t)}; D_i \right)$. To ensure non-negativity and highlight larger deviations more severely, we take the square of this value. This squared deviation measure $D_{i,m}$ approximates the square of the sum of gradients affecting the parameter evolution, indicating the significance of parameter updates on loss variability during the exploration phase. By squaring the sum of the gradients, we ensure that the deviation measure is always non-negative and that larger deviations are highlighted more severely than smaller ones.

The $C_{\exp}$ clients compile these deviations into a local matrix $G_{\text{local}}$, whose entries are the deviations for the model parameters. At each FL round, where only $K$ clients are involved, the server aggregates the $G_{\text{local}}$ matrices associated to those client to formulate $G_{\text{global}}$ through a normalization process:

$$G_{\text{global}} = \frac{1}{K} \sum_{k=1}^{K} \frac{G_{\text{local}_k} - \min(G_{\text{local}})}{\max(G_{\text{local}}) - \min(G_{\text{local}})} \qquad (3)$$

Here, the minimum and maximum values are taken over all $G_{\text{local}_k}$ matrices for $k = 1, \dots, K$. Each element of $G_{\text{global}}$ thus represents the mean normalized deviation for each parameter, scaled between 0 and 1. This process ensures that no single client's $G_{\text{local}}$ disproportionately influences $G_{\text{global}}$ due the possible presence of outliers in terms of deviations $D_{i,m}$. A value closer to 0 indicates minimal deviation, suggesting gradient stability during the exploration, hence scarce relevance of the parameter itself. Conversely, values near 1 highlight significant parameter deviations, pointing to more dynamic and potentially insightful areas of the loss landscape. Then, a binarization process is applied to $G_{\text{global}}$

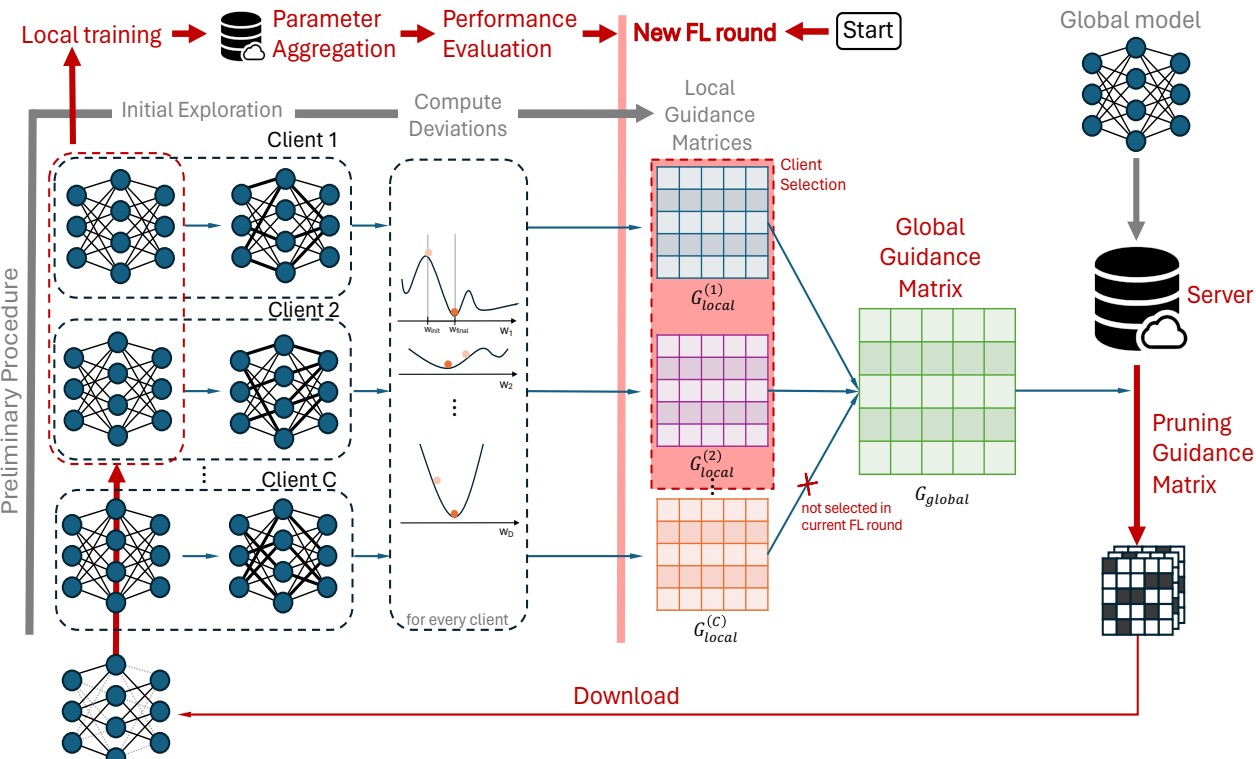

*Figure 1.* Illustration of the `AutoFLIP` pruning procedure. The local guidance matrices are computed a priori through the federated exploration phase. The global guidance matrix is computed by the server by aggregating the elements of the local guidance matrices corresponding to the clients participating in each FL round. The pruning mask is downloaded by the participant clients. All steps preliminary to the FL procedure are denoted in gray, while the steps intrinsic to the FL procedure with pruning are denoted in red.

where elements below $T_p$ are set to 0 and those above are set to 1:

$$PG_{\text{global},m} = \begin{cases} 0 & \text{if } G_{\text{global},m} < T_p \\ 1 & \text{otherwise} \end{cases} \quad (4)$$

The threshold $T_p$ directly determines the compression ratio of the model by setting the proportion of parameters to be pruned. Given their smaller influence, parameters corresponding to 0 are flagged for pruning, whereas those marked with 1 are retained, indicating important search directions within the model parameter space. During each FL round, the $K$ participating clients update $PG_{\text{global}}$ by incorporating their $G_{\text{local}}$ deviation values derived from the initial loss exploration phase.

To select an appropriate $T_p$, consider the desired compression ratio for the model. This ratio reflects the extent to which the model needs to be compressed while maintaining acceptable performance. By carefully selecting $T_p$ based on the desired compression ratio and empirical validation, we can achieve a well-balanced model that is both efficient and accurate, tailored to the specific needs of the FL task.

**The Proposed `AutoFLIP` Framework.** Here our aim is to argument how the parameter pruning mechanism based on loss exploration enters a general FL edge training framework. Algorithm 1 provides an overview of the entire framework of the proposed `AutoFLIP` algorithm for FL. It is composed by the following steps.

**Server initialization (Line 1).** The server is initialized with a global model that it is sent to all the clients. At this stage, the total number of clients undergoing exploration, the number of exploration epochs, and the pruning threshold are also decided.

**Exploration phase (Lines 2–3).**
The preliminary exploration phase aimed at understanding the relevance of each parameter (weight or bias) in view of loss improvement starts. For each client participating (in this study we select all the available clients), a local guidance matrix storing parameter deviations is computed.

**Mask update (Lines 5–7).** A FL round starts. The server selects $K$ clients that participate in the round. Only the local guidance matrices of those clients are considered to compute a global guidance matrix, which is then used to generate a binary mask for pruning. The mask contains ones

---

**Algorithm 1** `AutoFLIP` Algorithm

---

1: **Server Initialization:** Initial matrix $W_{\text{global}}^{(0)}$, number of clients for exploration $C_{\text{exp}}$, exploration epochs $E_{\text{exp}}$, pruning threshold $T_{\text{p}}$, FL rounds $R$, training epochs $E$, number of selected clients per round $K$
2: Server selects $C_{\text{exp}}$ clients for exploration
3: $G_{\text{local}_i} = (W_i^{\text{Initial}} - W_i^{\text{Final}})^2, \forall i \in [1, C_{\text{exp}}]$
4: **for** round $r = 1$ to $R$ **do**
5:     Server selects $K$ clients
6:     Compute $G_{\text{global}}^{(r)}$ using Eq. (3)
7:     Compute mask $PG_{\text{global}}^{(r)}$ using Eq. (4)
8:     **for** client $k = 1$ to $K$ **do**
9:         $W_{k,\text{pruned}}^{(r)} = W_k^{(r)} \odot PG_{\text{global}}^{(r)}$
10:         **for** each local epoch $e = 0$ to $E - 1$ **do**
11:             $W_{k,\text{pruned}}^{(e+1)} = W_{k,\text{pruned}}^{(e)} - \eta \nabla L_k\left(W_{k,\text{pruned}}^{(e)}\right)$
12:         **end for**
13:     **end for**
14:     $W_{\text{global}}^{(r)} = \frac{1}{K} \sum_{k=1}^{K} W_{k,\text{pruned}}^{(E)}$
15: **end for**=0

---

only for the parameters with normalized deviations higher than a prescribed threshold $T_p$.

**Pruning (Lines 8–9).** During each round, clients use the pruning mask to compress their models. This happens through element-wise multiplication between their weight matrix and $PG_{\text{global}}$ at that FL round. Parameters aligned with a 0 in $PG_{\text{global}}$ are pruned; those corresponding to a 1 are kept.

**FL round with reduced client models (Lines 10–14).** The standard algorithm FedAvg (McMahan et al., 2023) is used on the reduced framework. The pruned clients are trained. The server receives the local model updates and, upon aggregation, proceeds to update the global model with the FL aggregation strategy. Once updated, the global model is either ready for the next communication round or deemed ready for deployment if the convergence criteria are satisfied.

### 4.2. Robustness and Efficiency of **AutoFLIP**

Referring to (Fraboni et al., 2022; Fallah et al., 2020; Wang et al., 2023; Yin et al., 2024), we base our convergence guarantees on a federated stochastic aggregation scheme. The authors' assumptions on Lipschitz smoothness, convexity of local loss functions, unbiased gradient estimators, finite client answering times, and specific client aggregation weights form the theoretical backbone of `AutoFLIP`. These conditions ensure that the learning process remains stable and converges efficiently even in the presence of non-IID data distributions. With `AutoFLIP` each client experiences the same pruning strategy with $PG_{\text{global}}$, resulting in a substantial decrease in the variance ($\sigma_{\Delta W}^2$) previously de-

fined in Section 3.2, of weight updates for the global model. This uniform pruning strategy minimizes discrepancies in weight adjustments across clients by focusing updates on critical weights identified during the federated loss exploration phase. The reduction in variance helps to alleviate the bias caused by the non-IID setting, as shown in the work of (Zhu et al., 2021), thus promoting better global convergence.

Furthermore, (Yang et al., 2023; Redman et al., 2022; Tukan et al., 2022; Isik et al., 2022) provide a theoretical foundation for which pruned NNs can effectively learn signals. They demonstrate that pruning preserves the signal's magnitude in features and reduces noise, leading to improved generalization. These studies highlight that pruning, when done correctly, does not degrade the model's capacity to learn but rather focuses the learning on more relevant features. By focusing on parameters with significant contributions to the loss function, `AutoFLIP` ensures that the essential features are retained, thus maintaining or even enhancing model accuracy. As illustrated in Figure 2 for different NN topologies, the parameters in $G_{global}$ with minimal variability during the federated loss exploration phase are pruned, while those exhibiting high deviations, indicative of high gradients and significant contributions to the loss function variability, are retained. Note that higher frequencies are recorded for smaller deviation values, indicating that many parameters, according to our pruning strategy, are non important.

`AutoFLIP` enhances communication efficiency in FL by reducing model sizes transmitted between clients and the server, thus lowering bandwidth requirements for FL rounds. Its selective updating mechanism ensures only essential parameters, those significantly affecting model performance, are communicated. Integrating `AutoFLIP` into FL systems enhances faster training and inference times, lower energy consumption, and improved model scalability. The appendices D, E and F provide a detailed analysis of how `AutoFLIP` accelerates inference and improves training efficiency, demonstrating its significant role in lowering computational costs and boosting FL's overall efficiency.

## 5. Experiments

Inspired by (Hahn et al., 2022), we benchmark `AutoFLIP` across established datasets to evaluate its robustness in various non-IID environments. We explore three distinct partitioning approaches for creating strongly non-IID conditions: a **Pathological non-IID scenario**, which involves clients using data from two distinct classes, employing MNIST with a six-layer CNN (7,628,484 parameters) and CIFAR10 with EfficientNet-B3 (10,838,784 parameters), a **Dirichlet-based non-IID scenario**, which utilizes the Dirichlet distribution to distribute data among clients, with varying class counts

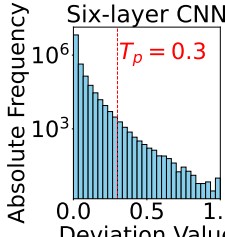 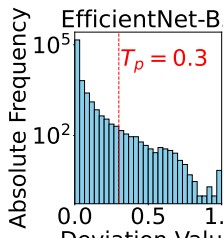 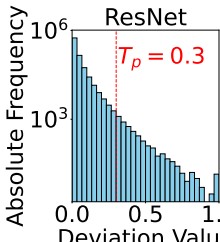 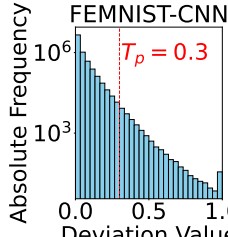 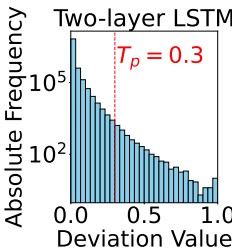

*Figure 2.* Distribution of parameter deviations in $G_{global}$ after exploration. Absolute frequency in log-scale is shown for each normalized deviation. Higher frequencies are recorded for smaller deviation values, indicating that many parameters are irrelevant for loss improvement.

per client, using CIFAR100 with ResNet (23,755,900 parameters), and a **LEAF non-IID scenario**, which adopts the LEAF benchmark (Caldas et al., 2019) with FEMNIST and Shakespeare datasets. For FEMNIST, a CNN architecture with 13,180,734 parameters is used. For Shakespeare, we consider a two-layer LSTM model with 5,040,000 parameters. Further details on these scenarios are provided in Appendix B.

### 5.1. Experimental Setup and Results

We evaluate `AutoFLIP` against both `FedAvg` without any model compression and with SotA FL pruning strategies, incorporating various parameter selection criteria: Random, L1, L2, Similarity, and BN mask, as described in (Wu et al., 2023). The experimental setup involves $C = 20$ (for LEAF non-IID scenario we employ $C = 730$ for Shakespeare and $C = 660$ for FEMNIST) clients, a batch size $B = 350$, and a learning rate $\eta = 0.0003$ over 200 total rounds $R$ with $K = 5$ (for LEAF non-IID scenario $K = 20$) clients selected per round. We incorporate a server momentum of 0.9 and use an SGD optimizer with weight decay. The exploration phase consists of up to $E_{exp} = 150$ epochs, and the pruning threshold is set to $T_p = 0.3$. Data is divided into 80% for training and 20% for testing, with global model performance assessed by the average prediction accuracy on the test sets. To ensure statistical validity, each experiment is repeated 10 times. We measure the compression rate to evaluate model size reduction and its impact. Experiments were conducted with an Intel Xeon X5680, 128 GB of DDR4 RAM, and an NVIDIA TITAN X GPU.

**Pathological non-IID)** Here, `AutoFLIP` achieves an average client compression rate of x1.74. At each round, we remove on average 3244298 parameters of the six-layer CNN for each participant client. For the EfficientNet-B3, we obtain an average compression rate of x2.1 with 5677458 deleted parameters. For a fair comparison with the baselines, we ensure that the number of parameters pruned matches

the compression ratio of AutoFLIP, quantified as 42% for the six-layer CNN and 52.38% for EfficientNet-B3.

The first two subplots in Figure 3 show the evolution of global model accuracy during the FL rounds for the four-layer CNN with the MNIST dataset and for EfficientNet-B3 with the CIFAR10 dataset. Refer to Appendix C for the evolution of the loss metric. In the case of the MNIST, the early rounds of FL show that `AutoFLIP` achieves slightly higher accuracy compared to both `FedAvg` and the other FL pruning strategies, among which RandomPruning emerges as the top performer. This indicates a faster convergence rate for our proposed method. However, the performance of the three baselines soon becomes comparable, with no clear superiority as the FL procedure progresses. We attribute this to the simplicity of the prediction tasks on the MNIST dataset compared to the excessive complexity of the four-layer CNN, which already possesses extremely good prediction capabilities that cannot be further enhanced by pruning. For the CIFAR10 dataset, we do not observe any advantage in using AutoFLIP over the other baselines. Surprisingly, all methods exhibit severe fluctuations in the accuracy convergence profiles up to FL round 100, after which they stabilize and become comparable.

**Dirichlet-based non-IID)** For ResNet, AutoFLIP achieves an average compression rate for the clients of x1.58, with 8,720,520 parameters pruned on average out of 23,755,900 total parameters. Hence, we adjust the percentage of parameters to be pruned to 36.71% for the different baselines.

The third subplot in Figure 3 illustrates the evolution of the global model accuracy during the FL rounds for ResNet on CIFAR100. Here, `AutoFLIP` exhibits a performance enhancement throughout the considered training rounds. At round 200, it achieves an accuracy of 0.987, compared to 0.918 for FedAvg and 0.925 for RandomPruning. This enhancement signifies the robustness of `AutoFLIP`, showcasing its ability to maintain elevated performance levels when integrated with larger-complex neural networks and

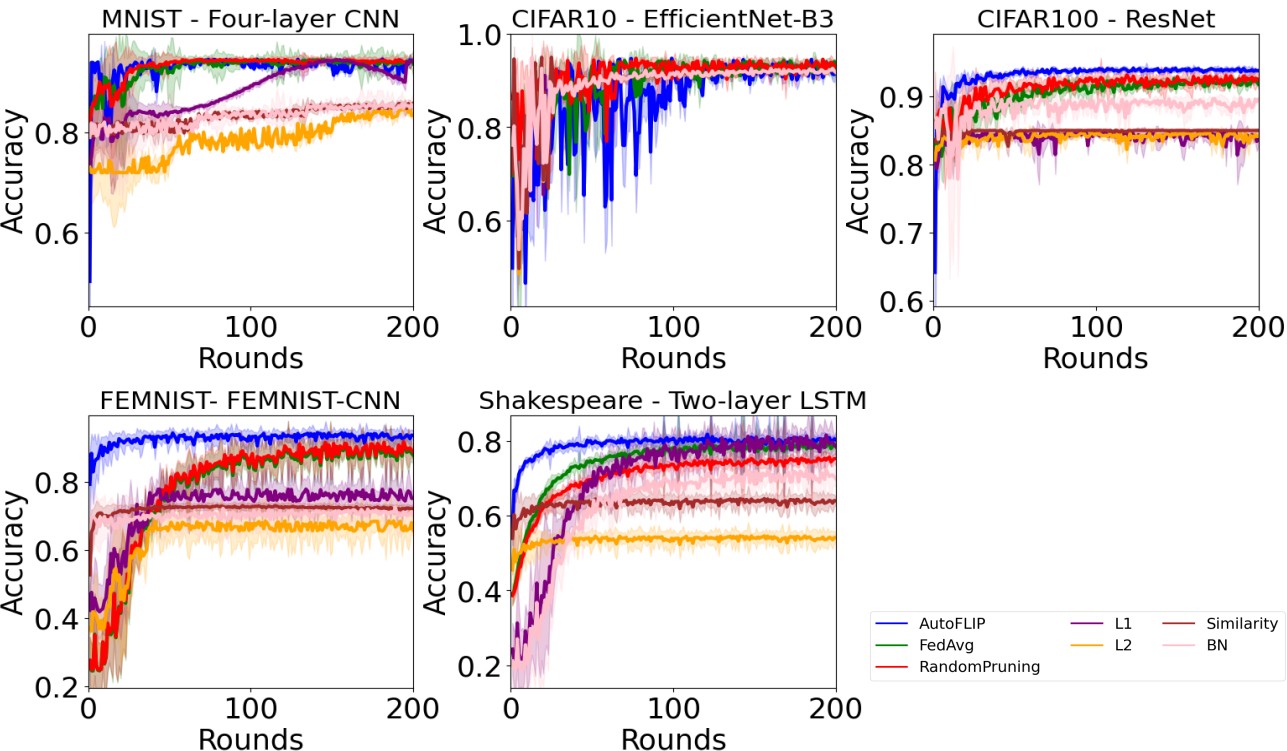

*Figure 3.* Average accuracy convergence profiles for the global model within the FL framework.

larger datasets.

**LEAF non-IID)** In this scenario, `AutoFLIP` achieves an average compression rate of x1.8 for 5858104 client parameters pruned out of 13180734. Hence, we adjust the number of parameters to be pruned for the different baselines to 44%. As observed in the last two subplots of Figure 3 for the FEMNIST and Shakespeare datasets, `AutoFLIP` consistently outperforms the other pruning strategies by a significant margin.

What stands out is the initial acceleration in convergence speed observed for `AutoFLIP`, firmly establishing it as a superior choice over `FedAvg`, RandomPruning and the other FL baselines. Furthermore, this superiority persists throughout the entire FL training procedure. The final average accuracy values are 0.985 for `AutoFLIP`, 0.905 for FedAvg, and 0.935 for RandomPruning on the FEMNIST dataset. For the Shakespeare dataset, the values are 0.815, 0.783, and 0.738, respectively. Here, even L1 proves to be competitive, reaching a final accuracy equal to 0.802. However, it demonstrates inferior initial convergence.

## 6. Conclusion and Limitations

We introduced `AutoFLIP`, an innovative automated federated learning (FL) approach that employs informed pruning to optimize deep learning (DL) models on clients with limited computational resources. Through extensive experiments across various non-IID scenarios, `AutoFLIP` has demonstrated its capacity to achieve high model accuracy and significantly reduce computational and communication overheads. It enhances convergence rates in federated settings and shows remarkable adaptability and scalability across diverse DL network architectures and multi-class datasets, particularly as the complexity of tasks increases.

**Limitations.** `AutoFLIP` shows promise but has limitations. It is primarily tested in the popular efficient single-server setting, not accounting for multi-server or hierarchical environments with diverse client capabilities and model structure. Our tests also assume standard conditions without data label noise.

**Future Research Directions.** `AutoFLIP` underscores its potential for future research avenues, such as leveraging loss exploration for guiding complex Neural Architecture Search (NAS) tasks. Enhancements will focus on refining `AutoFLIP`'s dynamic and adaptive pruning to better client

personalization. We aim to perform comparison analysis with other strategies from other domain such us like NAS or Client Dropout. Further, the impact on data privacy and defense against adversarial clients during the federated loss exploration phase has to assessed. Research will also explore the extension of `AutoFLIP` to more complex DL architectures and its integration into real-world applications across various domains such as healthcare and mobile computing.

**Broader Impact.** `AutoFLIP` enhances sustainability and efficiency in FL, reducing the energy footprint of training deep learning models. Its utility in sensitive sectors like healthcare and finance emphasizes its societal importance. However, deploying `AutoFLIP` requires careful consideration of ethical issues, including data privacy and biases. Proactive management and regulation are crucial to ensure its positive societal impact and responsible integration into critical fields.

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

# A. Ablation Study on $T_p$

We perform an ablation study to assess the sensitivity of our method to the pruning threshold parameter $T_p$. In particular, we check how the average accuracy and loss for the global model predictions vary for $T_p \in \{0.1, 0.2, 0.3, 0.4, 0.5\}$. We do this on two datasets: MINST in Figure 4 and CIFAR10 in Figure 5 from the Pathological non-IID scenario. In both cases, $T_p = 0.3$ seems the most convenient choice.

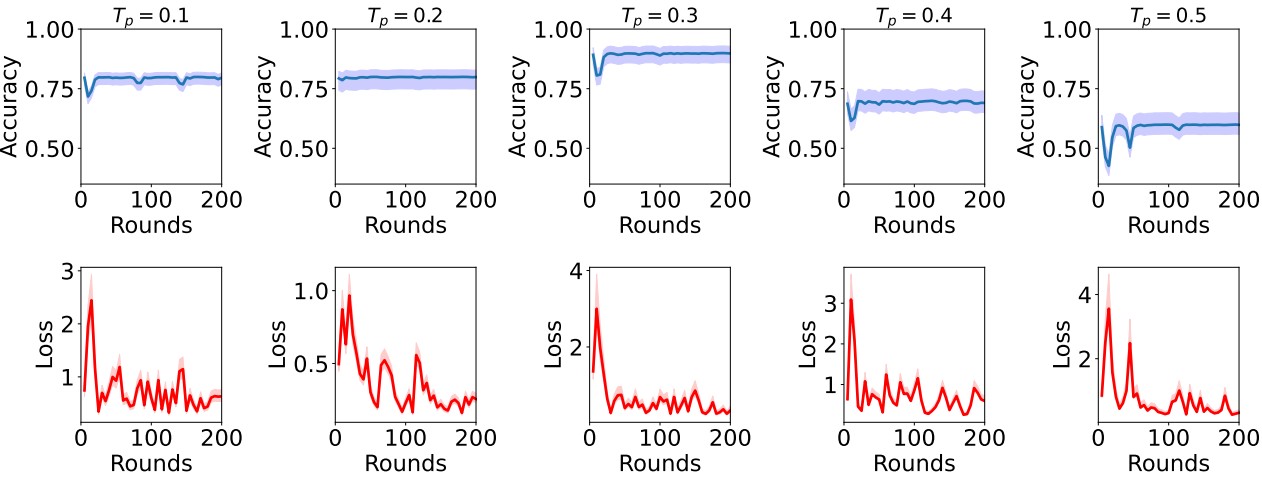

*Figure 4.* Ablation on $T_p$ for MINST/non-IID based on average accuracy (top) and loss (bottom).

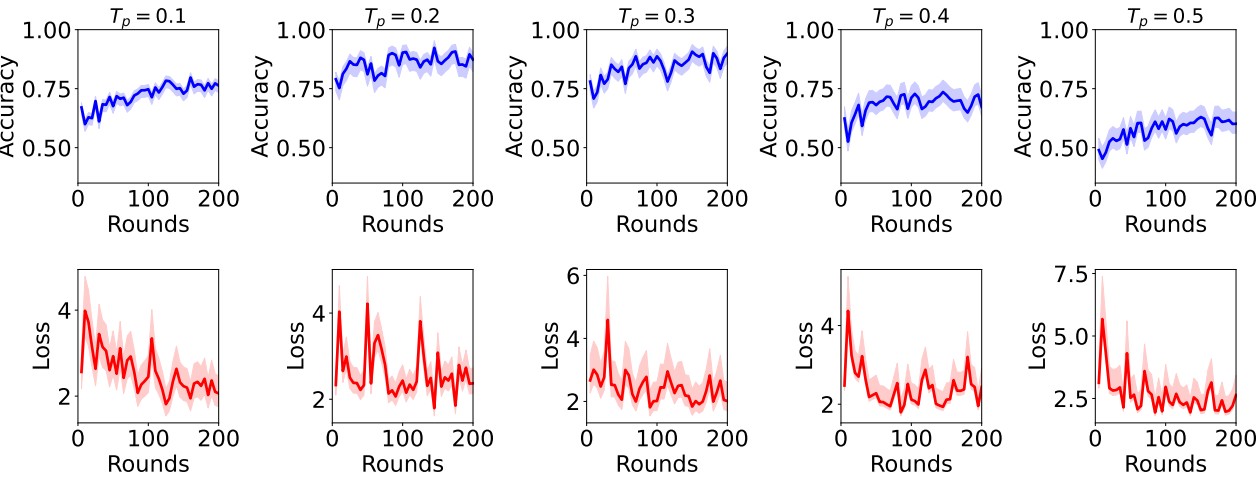

*Figure 5.* Ablation on $T_p$ for CIFAR10/non-IID based on average accuracy (top) and loss (bottom).

## B. Partitioning approaches

PATHOLOGICAL NON-IID

This experimental configuration is delineated by each client possessing data exclusively from two distinct classes within a broader multi-class dataset. Figure 6 illustrates this "pathological" data partitioning scenario within the CIFAR10 dataset across 20 clients. For our experiments, we select the MNIST dataset (Deng, 2012) with a six-layer CNN (7628484 parameters) and the CIFAR10 dataset (Krizhevsky, 2012) with EfficientNet-B3 architecture (10838784 parameters), following the guidelines in (McMahan et al., 2023) and (Tingting et al., 2023).

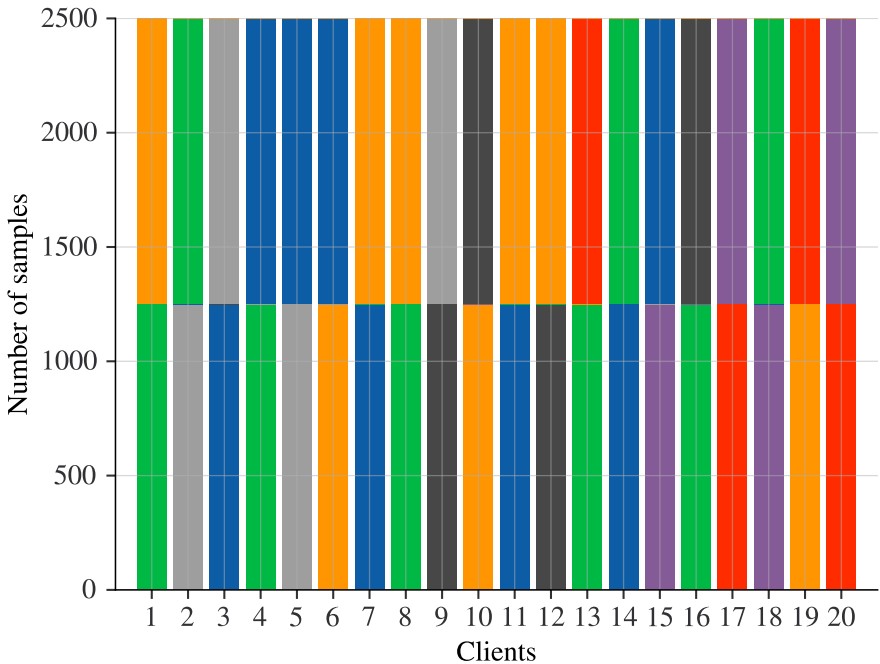

*Figure 6.* Illustration of pathological data partitioning on CIFAR10 for 20 clients, where each color represents a different class.

DIRICHLET-BASED NON-IID

This advanced experimental setup, as introduced by (Hsu et al., 2019), utilizes the Dirichlet distribution, modulated by a concentration parameter $\alpha$. Let $\boldsymbol{p} = (p_1, p_2, ..., p_N)$ be the class distribution for a given client, where $N$ is the number of classes. The Dirichlet distribution is defined as $\boldsymbol{p} \sim \text{Dir}(\alpha \cdot \mathbf{1}_N)$, where "Dir" denotes the Dirichlet distribution, $\alpha$ is the concentration parameter, and $\mathbf{1}_N$ is a N-dimensional vector of ones. In this context, a low value of $\alpha$, or $\alpha \to 0$, leads to distributions where most of the probability mass is concentrated on a single class, thereby indicating that each client's data is restricted to a single class. Conversely, as $\alpha \to \infty$, $\boldsymbol{p}$ approaches a uniform distribution, ensuring that the samples are evenly split across all clients. Figure 7 illustrates this "Dirichlet-based non-IID" data partitioning scenario within the CIFAR100 dataset across 20 clients, with individual colors denoting separate classes.

To address the complexities of larger datasets, we have extended our evaluation to include CIFAR100 (Krizhevsky, 2012) with a $\alpha = 100$, employing ResNet (23755900 parameters) (He et al., 2015) in alignment with the methodology proposed in (Hahn et al., 2022).

LEAF NON-IID

Utilizing the popular LEAF benchmark for FL (Caldas et al., 2019), we selected the FEMNIST and Shakespeare datasets to simulate closer real-world FL scenarios, with each dataset designed for specific tasks. The FEMNIST dataset is defined for a multi-class classification challenge involving 62 distinct classes. Conversely, the Shakespeare dataset is tailored for a next-character prediction task, requiring models to predict the subsequent character from a sequence of 80 characters,

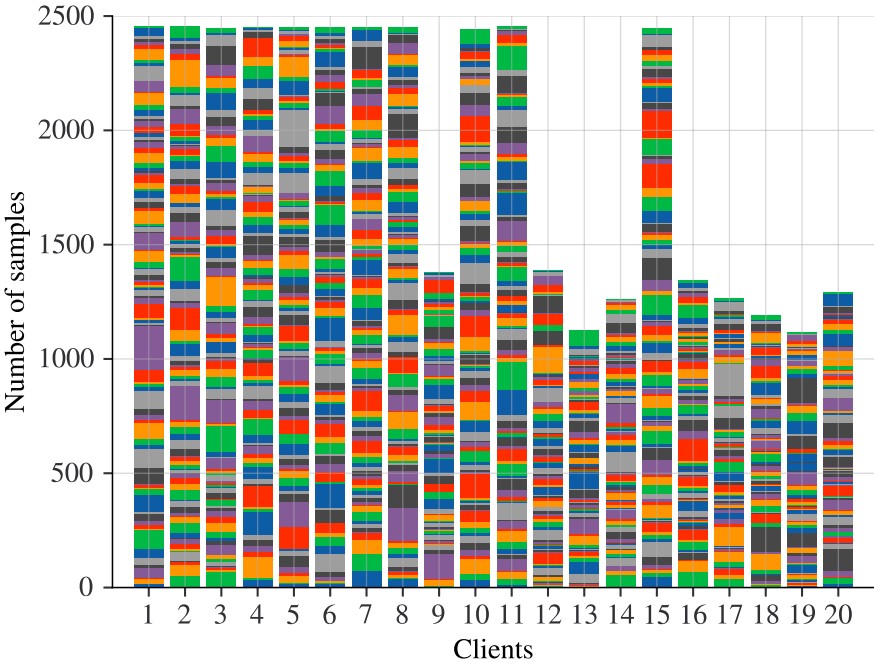

*Figure 7.* Illustration of Dirichlet-based non-IID data partitioning on CIFAR100 for 20 clients, where each color represents a different class.

thereby testing the model capabilities in sequential data processing and language modeling. The incorporation of the next-character prediction task allows for a comprehensive assessment of `AutoFLIP` adaptability and performance across diverse task types and deep neural network architectures, such as Long Short-Term Memory (LSTM) networks.

In our experimental setup, we employed the FEMNIST-CNN architecture, as delineated in (Caldas et al., 2019), for the FEMNIST dataset. For the Shakespeare dataset, we utilized a two-layer (LSTM) (5040000 parameters) model, in accordance with the specifications provided in (McMahan et al., 2023).

## C. Loss plots

We present in Figure 8 the loss convergence profiles for the global model participating in the FL procedure.for the global model. Here, we compare `AutoFLIP` to the different federated pruning strategies evaluated on both image recognition and text prediction tasks using five distinct datasets: MNIST, CIFAR10, CIFAR100, FEMNIST, and Shakespeare. Due to the varying complexities of each task, we use different model structures for different datasets.

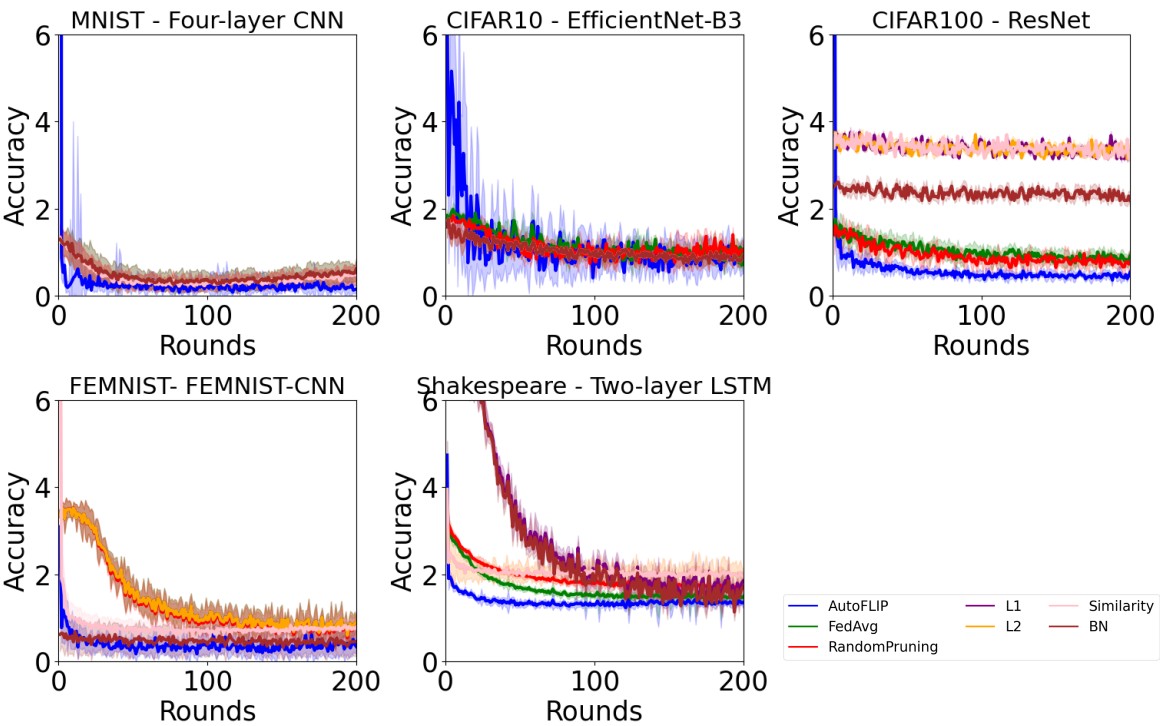

*Figure 8.* Average loss convergence profiles for the global model within the FL framework.

## D. Inference acceleration

In this section, we discuss the inference acceleration of `AutoFLIP`. When performing inference on the client's side with the pruned sub-model, we accelerate the inference time and reduce the computational consumption. Figure 9 shows the inference acceleration comparison after applying `AutoFLIP`. Notably, the FLOPs (floating point operations per second) in all the evaluated models are reduced. Table 1 shows that the Six-layer CNN deployed for the pathological non-IID experiment with MNIST, experienced a substantial decrease in computational load, equal to a 41.62% reduction in FLOPs. EfficientNet-B3, used for CIFAR10 in the pathological non-IID experiment, saw further improvements, reaching a FLOPs reduction of 46.44%. The deeper ResNet model, designed for CIFAR100 in the Dirichlet-based non-IID experiment, achieved a significant reduction in FLOPs, over 50%, highlighting the potential of `AutoFLIP` to streamline deep networks for more efficient inference. The FEMNIST-CNN and LSTM models, employed for the LEAF non-IID experiment, showcased a FLOPs reduction equal to 56.49% and 44.44%, respectively.

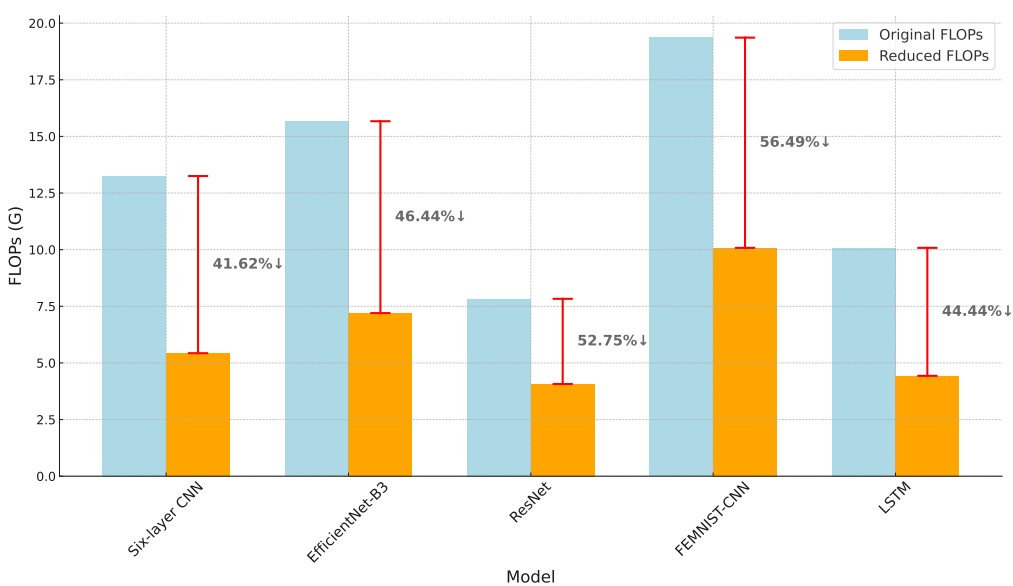

*Figure 9.* Original FLOPs and reduced FLOPs

*Table 1.* FLOPs comparison

| Model | Compression Rate | Original FLOPs | Reduced FLOPs | FLOPs % Reduced |
|---|---|---|---|---|
| Six-layer CNN | 1.74 | 13.25 G | 5.43 G | 41.62% ↓ |
| EfficientNet-B3 | 2.1 | 15.67 G | 7.20 G | 46.44% ↓ |
| ResNet | 1.58 | 7.83 G | 4.07 G | 52.75% ↓ |
| FEMNIST-CNN | 1.8 | 19.36 G | 10.08 G | 56.49% ↓ |
| LSTM | 1.8 | 10.08 G | 4.43 G | 44.44% ↓ |

# E. Training efficiency

To ascertain `AutoFLIP`'s impact on enhancing training efficiency within FL frameworks, we delve into an examination of the associated communication costs. For a practical perspective, the deployed models are trained to achieve a 90% accuracy threshold. As presented in (Yu et al., 2023b), the cost function employed for this evaluation is defined as:

$$\text{Cost} = \# \text{ Parameters} \times \# \text{ Rounds to Reach Target Accuracy} \times \# \text{ Clients} \times \text{Sample Rate}.$$

In Table 2, we observe the effectiveness of `AutoFLIP` in reducing communication costs across various non-IID scenarios with different models and datasets. Notably, the Six-layer CNN model, used in the MNIST dataset for the Pathological non-IID experiment, demonstrated a significant reduction in communication costs by 41.61%, which underscores `AutoFLIP`'s effectiveness in simpler architectures. This efficiency extends to more complex architectures, like EfficientNet-B3 and ResNet, employed for the CIFAR10 and CIFAR100 datasets respectively for the Dirichlet-based non-IID experiment, which also saw notable cost reductions of 30.93% and 29.88%. Similarly, the FEMNIST-CNN and LSTM models, used in the LEAF non-IID experiment, exhibited reductions in communication costs by 19.54% and 19.29%, respectively. These results highlight `AutoFLIP`'s broad applicability and substantial impact on training efficiency across a range of model complexities and dataset types.

*Table 2.* Comparison of the total communication costs

| Model | Rounds AutoFLIP | Rounds NoAutoFLIP | Cost AutoFLIP | Cost NoAutoFLIP | % Cost Reduced |
|---|---|---|---|---|---|
| Six-layer CNN | 3 | 58 | 189.45 GB | 324.43 GB | 41.61% ↓ |
| EfficientNet-B3 | 27 | 39 | 290.26 GB | 420.27 GB | 30.93% ↓ |
| ResNet | 7 | 49 | 712.70 GB | 1016.40 GB | 29.88% ↓ |
| FEMNIST-CNN | 280 | 348 | 369.06 GB | 458.69 GB | 19.54% ↓ |
| LSTM | 243 | 301 | 122.47 GB | 151.74 GB | 19.29% ↓ |

# F. Computation Cost

To evaluate `AutoFLIP`'s role in reducing computational effort, we investigate the number of parameters processed for a single client. Distinguishing between computational efforts on the global model and the clients is essential, with a particular focus on the client side. For `AutoFLIP`, each client handles a substantial number of parameters over an additional 150 exploration epochs ($E_{\text{exp}}$). From a practical standpoint, we compare `AutoFLIP` and `FedAvg` with RandomPruning with the same compression rate. The models are trained to meet a 90% of global accuracy. We define the computation cost function as:

$$\text{Computation cost for single client} = \text{Total Parameters Processed} \times \text{\# Epochs} \times \text{Sample Rate}$$

In Table 3, the pathological non-IID experiment with MNIST using the Six-layer CNN model shows a significant reduction in computational cost by 62.51%. This efficiency extends to more complex architectures like EfficientNet-B3 and ResNet, used for the CIFAR10 and CIFAR100 datasets respectively, with cost reductions of 46.41% and 58.22%. Similarly, the FEMNIST-CNN and LSTM models, employed in the LEAF non-IID experiment, demonstrated reductions in computational costs by 45.99% and 29.60% respectively. These results underline `AutoFLIP`'s broad applicability and substantial impact on reducing computational efforts across diverse model architectures and dataset types.

*Table 3.* Comparison of the total computation costs

| Model | Processed parameters AutoFLIP | Processed parameters NoAutoFLIP | % Cost Reduced |
|---|---|---|---|
| Six-layer CNN | 535,309,170 | 1,428,653,880 | 62.51% ↓ |
| EfficientNet-B3 | 2,005,175,040 | 3,740,891,680 | 46.41% ↓ |
| ResNet | 3,919,723,500 | 9,378,097,500 | 58.22% ↓ |
| FEMNIST-CNN | 15,303,653,440 | 28,338,578,100 | 45.99% ↓ |
| LSTM | 5,871,600,000 | 8,335,200,000 | 29.60% ↓ |

