# OpenReview forum: "Adaptive Model Pruning in Federated Learning through Loss Exploration"
_ICML.cc/2024/Workshop/WANT — WANT@ICML 2024 Poster_

### Official Review · Reviewer_mPok · 2024-06-07
**The main idea is clear and natural but the paper needs improvement.**

**Confidence:** 4

**Summary:**

The paper proposes a model pruning approach in federated learning to combat communication overhead and the effects of non-IID datasets across clients. They achieved this by conducting an exploration phase before performing federated learning, where the impact of each parameter on each client's dataset is determined. They demonstrated the effectiveness of their method through experiments.

**Strengths:**

The idea of determining the importance of each parameter for each client based on the local data seems like a natural solution.
The paper's main idea is explained well.

**Weaknesses:**

There are some writing problems. The Notation section is repeated in the next section, and the line numbers for the algorithms are all zero. Equation 3 also has notation issues.

The objective is not clearly expressed, whether the pruning is intended to combat non-IID data or improve communication efficiency. The authors claim multiple times that their objective is to minimize variance through uniform pruning but provide no proof that this method minimizes variance during federated learning.

There are ambiguities without explanations, such as E_exp = 150. Additionally, there are a few new parameters like E_exp, C_exp, and T_p for which no guidance is provided on how to choose them.

---

### Meta-Review · Area_Chair_QVFK · 2024-06-18

**Recommendation:** Accept (Poster)
**Confidence:** 3

**Metareview:**

This submission introduces an adaptive pruning strategy for the federated learning setting; the technique involves computing weight importance during the FL exploration phase. Reviewer sentiment for this submission appears positive overall - the idea appears to be novel and has been explained well. However, writing (especially notation) could be improved and the objectives more clearly stated. I recommend acceptance (poster).

---

### Decision · Program_Chairs · 2024-06-18

**Decision:**

Accept (Poster)

**Comment:**

We thank the authors for their time and contribution to WANT and we are pleased to share that after the reviewing process the paper has been accepted. Congratulations! We encourage the authors to consider reviewers' feedback for the improvement of the camera-ready version. We hope to see you in person at the workshop and brainstorm on efficient training research together!